# Ultrasonic Surgical Aspirator in Intramedullary Spinal Cord Tumours Treatment: A Simulation Study of Vibration and Temperature Field

**DOI:** 10.3390/bioengineering12080842

**Published:** 2025-08-04

**Authors:** Ludovica Apa, Mauro Palmieri, Pietro Familiari, Emanuele Rizzuto, Zaccaria Del Prete

**Affiliations:** 1Department of Mechanical and Aerospace Engineering, University of Rome La Sapienza, Via Eudossiana 18, 00184 Rome, Italy; emanuele.rizzuto@uniroma1.it (E.R.); zaccaria.delprete@uniroma1.it (Z.D.P.); 2Department of Human Neurosciences, Division of Neurosurgery, Policlinico Umberto I University Hospital, Sapienza University of Rome, 00157 Rome, Italy; mauro.palmieri@uniroma1.it (M.P.); pietro.familiari@uniroma1.it (P.F.)

**Keywords:** Intramedullary Spinal Cord Tumors (IMSCTs), microsurgical resection, Cavitron Ultrasonic Surgical Aspirator (CUSA), Finite Element Analysis (FEA), strain and temperature measurements, human spinal cord model

## Abstract

The aim of this work is to analyse the effectiveness of the medical use of the Cavitron Ultrasonic Surgical Aspirator (CUSA) in microsurgical treatment of Intramedullary Spinal Cord Tumors (IMSCTs), with a focus on the thermo-mechanical effects on neighbouring tissues to assess any potential damage. Indeed, CUSA emerges as an innovative solution, minimally invasive tumor excision technique, enabling controlled and focused operations. This study employs a Finite Element Analysis (FEA) to simulate the vibratory and thermal interactions occurring during CUSA application. A computational model of a vertebral column segment affected by an IMSCT was developed and analysed using ANSYS 2024 software. The simulations examined strain distribution, heat generation, and temperature propagation within the biological tissues. The FEA results demonstrate that the vibratory-induced strain remains highly localised to the application site, and thermal effects, though measurable, do not exceed the critical safety threshold of 46 °C established in the literature. These findings suggest that CUSA can be safely used within defined operational parameters, provided that energy settings and exposure times are carefully managed to mitigate excessive thermal accumulation. These conclusions contribute to the understanding of the thermo-mechanical interactions in ultrasonic tumour resection and aim to assist medical professionals in optimising surgical protocols.

## 1. Introduction

Intramedullary Spinal Cord Tumors (IMSCTs) represent a relatively rare but clinically significant group of central nervous system neoplasms that can affect both pediatric and adult populations [1,2]. These tumors are predominantly primary in origin, arising within the spinal cord parenchyma, while metastatic involvement of the intramedullary space is more rare [1]. Depending on the type of spinal cell affected, it can be distinguished in ependymoma and astrocytoma [3]. The distinction between these two tumor types is not only histopathological but also has relevant surgical implications. Ependymomas typically exhibit well-defined margins, which makes them more tractable to complete resection. In contrast, astrocytomas tend to infiltrate surrounding neural tissue, requiring removal up to the point where a visible interface with healthy white matter is observed. As a matter of fact, astrocytomas are more challenging to be resected without causing neurological damage [4].

Microsurgical resection remains the gold standard treatment for IMSCTs, aiming to achieve maximal tumour removal while preserving neurological function [5]. Radiotherapy and chemotherapy are generally considered secondary options, reserved for high-grade infiltrative tumours, recurrences, or cases in which surgery is contraindicated due to anatomical complexity or patient-specific risks [6]. Despite the continuous advancement of intraoperative tools and imaging techniques, such as neuro-navigation, intraoperative Magnetic Resonance Imaging (MRI) and neurophysiological monitoring, the surgical management of IMSCTs remains technically demanding. This procedure requires meticulous dissection and exceptional precision to avoid injury to critical spinal cord structures, highlighting the need for experienced surgical teams and a tailored, patient-specific approach.

The Cavitron Ultrasonic Surgical Aspirator (CUSA) emerges as an innovative solution, claiming to be a minimally invasive technique of tumour excision and allowing for controlled and focused surgical operations [7,8]. It was originally developed for dental plaque removal and later adapted for neurosurgical applications in the 1970s [9]. The CUSA EXcel system (Integra LifeSciences, Princeton, NJ, USA) is among the most commonly employed devices in microsurgical resection of IMSCTs. It consists of a console unit and a handpiece, which is composed by a transducer of ultrasonic vibration and a disposable surgical tip that takes advantage of low ultrasonic frequency vibrations to dissect and denature tissues with low fibre content, by enforcing mechanical energy through pulses. It integrates aspiration and irrigation to remove debris and cool the tip, minimizing thermal injury [10]. The handpieces of ultrasonic aspirators can use two technologies to convert electrical energy into mechanical ultrasonic waves. The first one is electrostriction [11], which is based on piezoelectric ceramic crystals, which react very quickly to electrical field applications to produce alternating longitudinal vibration, thus causing mechanical deformation in dielectric materials. The second technology is magnetostriction [11] which results in a change in the shape of magnetic materials during the magnetisation process. Magnetostriction, in particular, is actually the most widely employed technique for tumour resection, and one of the most representative instruments employing this technology is the CUSA EXcel. In the CUSA a nickel alloy is immersed in a magnetic field created by electrical current in a coil and the alternate current signal generated in the console unit causes the transducer to expand and contract. This motion is then amplified by the handpiece and the tip, so that the amplitude range is on the order of hundreds of μm. The main disadvantage of this technology is that the handpiece must be cooled to prevent overheating. In this regard, an irrigation fluid flows coaxially around the tip with the double scope of keeping it cool and suspending fragmented tumour tissue in solution, to minimise blockage. The fluid is then aspirated through pre-aspiration holes before it reaches the end of the tip, where an encircled silicone flue is allocated to guarantee a continuous pathway for fluid delivery.

The surgical experience emphasizes that, as a result of irrigation, it is possible to prevent thermal injury to surrounding tissues. However, challenges persist regarding the effects of ultrasonic vibration and residual temperature changes on adjacent tissues during microsurgical resection of Intramedullary Spinal Cord Tumours (IMSCTs). These factors are critical when operating near delicate neural or vascular structures, where even minor alterations in the temperature can impact procedural safety and outcomes. Indeed, while these instruments (including CUSA) integrate advanced irrigation and aspiration mechanisms to mitigate risks, the precise interaction between ultrasonic vibrations, temperature fluctuations and surrounding tissues are still poorly understood.

Current research focuses primarily on the clinical outcomes of surgical treatments [12,13], with few studies investigating the thermal and mechanical effects of the working instrument. Suetsuna et al. [12] examined surgical treatments on animals, using spinal cord evoked potentials (SCEP). This technique involved placing electrodes in proximity to the spinal cord to measure electrical activity in response to a stimulus. In this manner, it was possible to monitor the integrity of sensory and motor pathways during surgical procedures and diagnose spinal cord disorders. In the experiment, canine subjects were used. The energy of the ultrasonic aspirator ranged from 0 to 100% (maximum 300 μm) and the pressure of suction was maintained constant at 400 mm/Hg. The results showed that, with 80% of energy, the functionality of the spinal cord was electrophysiologically damaged. Changes in the SCEP signals were observed in 10 s tests with 80% or more energy and in 20 s tests with 60% or more energy. The study concludes that for treatment with 60% energy or below and with an activation time at one point of less than 10 s, the ultrasonic surgical aspirator could safely be used on the Dura Mater without rupturing the tissue or damaging the spinal cord. Young et al. [13] employed the same methodology of potential measurement to evaluate the effects of CUSA on nervous tissue, in particular on rat sciatic nerve, detecting the abolishment of the action potential when the vibrating tip touched the nerve. They also performed resections of small portions of the spinal cord with the CUSA on cats, producing localised lesions and observing no severe impediment in action potential conduction. These studies suggested that the effects of the ultrasonic vibration are relatively localised, even if there is still a lack in the evaluation of the effect of the Cavitron ultrasonic aspirator for the removal of spinal cord astrocytomas.

CUSA has been in clinical use since the 1980s and modern, more advanced versions, such as the CUSA by Integra and the Sonopet by Stryker, are now widely employed, particularly in spinal surgery involving bone and soft tissue resection. However, it is important to remark that their application to Intramedullary Spinal Cord Tumours remains extremely limited for different reasons. Firstly, not all neurosurgical centers frequently treat spinal intradural or intramedullary tumours due to their relative rarity and complexity. Consequently, there are not established guidelines, recommendations, or large clinical series specifically addressing the use of ultrasonic aspirators in Intramedullary Spinal Cord Tumour surgery. Then, despite 40 years of CUSA use, clinical practice in this specific context remains careful due to the absence of clear evidence on its safety profile for neural tissue. The few existing studies are outdated and insufficiently comprehensive.

In this context, while the use of CUSA in broader spinal surgery is well-known, its direct and safe application within the delicate environment of the spinal cord parenchyma remains under-investigated. Our study aimed precisely to fill this gap in the literature about the interaction of temperature and vibration effects over the site of application and the surrounding regions. In this study we focus on simulating the effects of CUSA ultrasonic vibrations and temperature variations on tissues adjacent to the tumour. By exploring these dynamics, we aim to enhance the safety and efficacy of CUSA-based procedures, contributing to the development of optimized, patient-specific surgical strategies for IMSCT management.

## 2. Materials and Methods

### 2.1. Model Construction

Computer-aided design (CAD) of the human spinal cord was designed and assembled in Autodesk Fusion 360 2024 (Autodesk Inc., San Rafael, CA, USA), and then imported in Ansys 2024 (ANSYS Inc., Canonsburg, PA, USA) for segmentation and mesh generation. Since the cervical spine is the most common location (33%) for intramidullary spinal cord tumours, followed by the thoracic region (26%) and the lumbar region (24%) [14,15], we modelled the segment of the vertebral column corresponding to the cervical tract, which includes vertebrae from C3 to C7. Moreover, to simplify the structure, we disregarded the curvature of the column and assigned a cylindrical coaxial structure to the elements within the cavity. The structural details of the human cervical model are illustrated in Figure 1. The figure provides multiple perspectives of the model and a sagittal cross-sectional view integrating Intramedullary Spinal Cord Tumor modeling (outlined in red in Figure 1e).

The model of the human spinal cord is composed of five vertebrae, which are the primary structural elements. Each vertebra consists of a cylindrical body and a bony arch that protects the spinal cord. Between adjacent vertebrae lie the intervertebral discs, which are cartilaginous pads that act as cushions to absorb shocks while ensuring flexibility without compromising the supportive strength of the vertebral column. The outermost layer is the epidural fat, which fills the cavity, providing additional cushioning and protecting the spinal cord and nerve roots. Surrounding the spinal cord is the dura mater, a tough and fibrous membrane that offers essential protection. Within the meninges circulates the cerebrospinal fluid (CSF), a clear liquid that cushions the spinal cord, removes waste, and supplies vital nutrients. At the core of the system lies the spinal cord, a long, slender bundle of nervous tissue extending from the brainstem down the vertebral column, acting as the primary pathway for transmitting nerve signals between the brain and the rest of the body [16,17]. Finally, as reported in [18], IMSCTs do not exhibit significant discontinuities in mechanical properties with respect to the surrounding spinal cord tissue. This assumption is further supported by tissue mechanical characteristics, which will be presented in the following sections, indicating that the tumour’s properties are comparable to those of the spinal cord. Based on this, the tumour has been modelled as a smoothed cylinder [19,20,21] embedded within the spinal cord, located near the C4 and C5 vertebrae, as shown in Figure 1e. Table 1 shows the geometrical dimensions set for each compartment of the model.

### 2.2. Finite Element Analysis (FEA)

A Finite Element Analysis (FEA) was performed in Ansys 2024 (ANSYS Inc., Canonsburg, PA, USA). Three different analyses coupled as shown in Figure 2 are needed for a complete evaluation of thermal and vibration effects.

The simulation procedure starts with the harmonic response, in which the ultrasonic input is modelled as a sinusoidal displacement to assess the structural response to vibrations. Subsequently, heat generation is estimated based on the dissipated energy and is incorporated into the coupled field transient analysis, which elucidates the associated transitory behaviour. Finally, the steady state thermal analysis concludes the study by providing insights into the heat flux and temperature field when stationarity is achieved.

In particular, the simulation is based on the application of the CUSA, which has two available handpieces differing for working mode, namely 23 kHz and 36 kHz modes. As regards the operating amplitudes, the transducer can expand and contract up to 35 μm, and the handpiece and the tip then amplify this motion up to 10-fold. This provides a range of amplitudes, increasing the fragmentation power of the system through the control panel (0–100%), of 35 to 355 μm [9]. The 36 kHz-handpiece supports various disposable tips, including the smallest PrecisionTip (1.14 mm diameter) and the standard tip (1.98 mm diameter), available in different lengths and shapes (straight or curved). As a result, weakly bonded tissues like tumours fragment easily, while stronger tissues like nerves resist. The CUSA EXcel offers suction up to 600 mmHg for effective coupling and tissue fragmentation.

Table 2 shows the mechanical and thermal properties of the model components at the physiological human body’s reference temperature of 37 °C. At this point, it has to be remarked that a series of assumptions have been taken into account in the simulation. First, all materials were considered homogeneous and isotropic [22,23,24]. This implies, for example, the disgregation of the dual bone composition, namely trabecular bone and cortical bone. Similarly, the dual composition of the intervertebral disc, consisting of a ’nucleus pulposus’ at the center and an ’annulus fibrosus’ externally, is also ignored. Same considerations are valid for the spinal cord, where we did not consider the distinction between white matter and grey matter. Another simplification concerns the cerebrospinal fluid, which has been treated as a solid component with a very low Young’s Modulus, despite its liquid nature, as suggested by [25,26].

Figure 3a shows the mesh geometry generated by an adaptive tethraedal mesh algorithm included in Ansys. The final model size resulting from multiple mesh densities refinement contained approximately 49,916 tetrahedral elements for the full anatomy of spinal cord model. For body sizing, a sphere of influence with a radius of 20 mm was placed at the centre of the tumour, with the element size set to 1 mm, as illustrated in Figure 3b. To simulate the effect of the sonicator, a circular face with a diameter of 2 mm was created on the tumour to represent the spot of application of the device (Figure 3c). This face was sized at 0.2 mm.

### 2.3. Model Boundary and Initial Conditions

Harmonic boundary conditions were applied to replicate the load exerted on the tumour. Specifically, a sinusoidal displacement was defined on a circular surface with a diameter of 2 mm, which represent the tip of the device. Two simulation tests were performed by varying the peak-to-peak amplitude of the displacement signal, i.e., 70 μm and 213 μm, with a frequency of 36 kHz for both tests. These parameters, corresponding to 20% and 60% of CUSA power, were selected as they represent common low and medium power settings, respectively, typically employed in clinical treatments [6,9]. The selected surface corresponds to the contact interface between the device and the tumour, accurately mimicking the mechanical interaction during treatment. The input displacement lies along the y axis and is thus applied orthogonally to the tumour’s surface, simulating the prone position of the patient during the treatment. The fixed support was applied on the outer vertebrae to simulate anchoring.

As regard to the thermal behaviour, the estimation of heat flux is obtained from the harmonic response and it is used to set up the thermal analyses. The calculations to arrive at this estimate is made through equivalent radiated power method. In detail, the equivalent radiated power (*Q*) represents the energy emitted due to harmonic vibration in the specific application area (*A*). Assuming that 100% of this energy is dissipated as heat, the resultant heat flux through the tumour’s surface (*q*) can be estimated as follows:(1)q=QA

The heat flux is also applied to the circular surface corresponding to the tip of the device. The heat is expected to propagate through both conduction and convection, as a result of vibratory activity, which could cause tissue detachment during application. Therefore, a convection condition is also assumed, considering a reasonable average value for air, as suggested in literature [38].

In the transient study, the total time of application in the analysis settings has been set to 1 s. This is because it is reasonable to assume that, even in the cases in which the treatment lasts longer (on the order of tens of seconds), the tip does not remain stationary at one point but moves around, involving adjacent zones. Thus it is possible to consider this time of application as a maximum, for the sake of safety. Moreover, the effects of the irrigation fluid from the handpiece, which is expected to cool down the stimulated tissue, have not been taken into account in order to hypothesise a worst-case scenario.

## 3. Results

### 3.1. Strain Fields

Figure 4 shows the strain field distributions on the tumour during the simulation test when applying a sinusoidal displacement of 70 μm. It is possible to note that the tumour is subjected to local strain with a peak in the y direction of 5.02 ɛ due to its very low Young Modulus’ value (1.8 Pa). The effects of the applied strain are extremely localised at the site of application of the device, with a spreading of about 4 mm of diameter across the neiborhood tissues, as shown in Figure 4. As a consequence, the other components of the human cervical model basically show no involvement.

Figure 5 shows the strain field distributions on the tumour during the simulation test when applying a sinusoidal displacement of 213 μm. It is interesting to note that the tumour is again subjected only to very local strain and the effects remain circumscribed in approximately the same diameter of 4 mm. However, in this case there is a significant increase of the peak of strain, equal to 15.29 ɛ, which may correlate with potential damage. Again, this behaviour might be due to the very low value of the Young Modulus, that cause the penetration to proceed deep without extending radially its impact. As a result, the surrounding components of the model remained unaffected by the ultrasonic action, indicating a highly localized mechanical interaction. Indeed, tissues with weak cellular adhesion and low fibre content, such as certain tumour types (e.g., glioblastomas, high-grade astrocytomas, or metastatic carcinoma), tend to fragment more easily, in contrast to structures like nerves, which exhibit strong intracellular cohesion. Notably, this behavior is commonly associated with astrocytomas, which are often characterized by a soft, infiltrative architecture that makes them particularly susceptible to mechanical disintegration under ultrasonic action. In contrast, other tumour types, such as ependymomas, may present firmer consistency, greater tissue cohesion, and more defined structural boundaries, often requiring sharp dissection rather than fragmentation.

### 3.2. Heat Generation Estimation

Figure 6 shows the equivalent radiated power obtained from the harmonic response when a displacement signal of 70 μm was applied. The value corresponding to 36 kHz of frequency is *Q* = 0.3907 W. The surface of the tumour (*A*) was derived from Ansys geometry and materials and it is equal to 6.5742 × 10^−4^ m^2^. From Equation (1) we obtained the heat flux across the entire surface of the tumour due to the dissipation of emitted energy equal to *q* = 600.94 W/m^2^. This value was then applied as a thermal forcing to investigate the thermal behaviour.

Figure 7 shows the equivalent radiated power obtained from the harmonic response when a displacement signal of 213 μm was applied. In this case, the value corresponding to 36 kHz of frequency is *Q* = 3.6579 W and the heat flux was again obtained through Equation (1) considering the same value of the tumour surface. The resulting heat flux across the entire surface of the tumour due to the dissipation of emitted energy was equal to *q* = 5567.6 W/m^2^, that was therefore imposed as a thermal forcing to investigate the thermal behaviour.

### 3.3. Temperature Fields

The vibratory phenomenon induces a heat flux that simulates the effects of thermal energy transfer within the tissues, which causes an increase in the temperature of the adjacent structures. Upon reaching stationarity, the area of application of instrument’s tip experiences a maximum temperature of T = 37.59 °C, as shown in Figure 8a, due to the dissipation of vibratory energy. This effect is gradually perceived up to the vertebrae in the region situated near the tumour (Figure 8b). The spatial temperature field decreases in a similar manner with increasing spherical radial distance, considering that the tumour, spinal cord, and CSF have roughly the same thermal conductivity, as shown in Table 2. Bone is also involved, but with a substantial reduction of intensity. The heat flux is effectively reduced when it encounters epidural fat, as shown in Figure 8c. Figure 8d shows a view obtained by cutting the model with a plane orthogonal to z axis at the point where the load is applied. Hence it can be observed the radial propagation of the heat flux through the different layers. In particular it can be noticed the important reduction reported for crossing the epidural fat, which lead to a very small area involved in the bone, with respect to the diffusion on the side of the tumour itself and the spinal cord.

The vibratory phenomenon in the second simulation induces a more substantial heat flux, reflecting the effects of increased thermal energy transfer within the tissues. This enhancement is due to the higher power used, leading to a more pronounced temperature increase in the adjacent structures. Upon reaching stationarity, the area of application exhibits a maximum temperature of T = 42.48 °C, significantly higher compared to the previous simulation (Figure 9a). This effect is similarly perceived up to the structures located near the tumour. The spatial temperature field decreases with increasing spherical radial distance, maintaining a similar pattern to the first simulation due to the comparable thermal conductivity of the tumour, spinal cord, and CSF, as shown in Figure 9b,c. When the intensified heat flux encounters the epidural fat, it is significantly mitigated, as evident in Figure 9d. The bone’s involvement is also more pronounced, although the intensity of heat transfer still shows a notable reduction. The cross-section view (Figure 9d) enables a clear visualization of the radial propagation of the heat flux through the various tissue layers, which exhibits the same pattern but with increased intensity in this second simulation. It is particularly noteworthy how the heat flux encounters significant attenuation as it crosses the epidural fat layer. This attenuation markedly restricts the area affected in the bone, in stark contrast to the more extensive spread of heat on the side of the tumour itself and the spinal cord.

Despite the higher overall heat flux in this scenario, the epidural fat continues to serve as an effective thermal barrier, thus preventing extensive thermal penetration into the bone. While this low temperature increase is unlikely to have significant clinical impact, since minor increases in bone temperature generally do not compromise structural integrity, it remains crucial to report and monitor such changes, particularly in areas adjacent to sensitive neural structures. This is especially relevant in anatomically constrained regions or in cases of repeated ultrasonic application, whereas localized thermal effects could become more significant. On the other hand, the regions adjacent to the tumour and the spinal cord display a more pronounced thermal influence, reflecting the higher energy input of the simulation. The differential in heat distribution underscores the role of anatomical structures in modulating the effects of thermal therapies, with implications for therapeutic efficacy and safety.

Table 3 shows the maximum temperature values reached on each tissue during the simulation tests performed with a sinusoidal displacement signal equal to 70 μm and 213 μm. In the area of application on the tumour, where the maximum value of temperature was recorded, the simulation test at 213 μm of displacement resulted in a temperature increase of 4.89 °C compared to the test at 70 μm of displacement. While tissues such as bone and the intervertebral disc exhibited minimal temperature variations in both tests and only slight differences between them, other tissues demonstrated more pronounced temperature differences between the two simulation tests.

## 4. Discussions and Model Limitations

The aim of this work was to assess the effectiveness of the medical use of the CUSA in microsurgical treatment of Intramedullary Spinal Cord Tumors (IMSCTs). Our specific focus was on understanding the thermo-mechanical effects on adjacent tissues to identify any potential damage. To achieve this, we utilized Finite Element Analysis to simulate vibratory and thermal interactions during CUSA application. A computational model of a vertebral column segment containing an IMSCT was developed and analyzed using ANSYS software. The simulations investigated strain distribution, heat generation, and temperature propagation within the biological tissues across two distinct simulation tests, varying the peak-to-peak amplitude of the displacement signal, i.e., 70 μm and 213 μm, with a frequency of 36 kHz for both tests.

As regards strain, for both the simulation tests the results showed an effect mostly localized at the site of application of the device, covering a diameter of approximately 4 mm of involved tissue. While the other components of the human cervical model show no strain. On the other hand, as regards temperature in the first FEA results, the model does not show a significant increase overall. The maximum increase is about 0.6 °C. This result is consistent with existing literature. For instance, Zannou et al. [23] measured the temperature field as a result of low-frequency ultrasonic stimulation (around 20 kHz) of the spinal cord. To this, they employed a Spinal Cord Stimulation (SCS) waveform power, which is an electrical stimulation that warms the tissue through Joule effect and developed a similar FEM model. Results showed that the temperature at the dorsal spinal cord increased by 0.18–1.72 °C during 3.5 mA of stimulation amplitude at 10 kHz. Furthermore, our FEA predictions confirmed the observations of Suetsuna et al. in [12], showing that the effects of CUSA treatment revealed localized responses and did not report significant damage for treatments with 60% energy or below (the amplitude of 70 μm used in the input falls within this range, and the limit is exactly 213 μm) and an activation time at one point of less than 10 s.

Focusing on temperature, since hyperthermia is currently being employed to treat malignant neoplasms [39] in many anatomic sites and so this collateral effect could bring benefits to the therapy, the interest is in the consequences in spinal cord. In particular its maximum tolerable temperature must be known, also because it may be hard to warm selectively the tumor alone. The study of Seiji Uchiyama et al. in [40] can be useful in this investigation. They performed a heat application using radio-wave electrodes placed in close contact with the surface of the abdomen and the back of canine subjects. The output power and heating time were adjusted according to the monitored temperature and were generally about 200–400 W, 10–20 min and with a radio frequency energy at 13.9 MHz. To avoid inserting temperature sensors into the spinal cord, a thermocouple sensor was placed in dorsal and epidural space. The results show that after a brief application period during which energy is applied to heat up (less than 1 min), the temperature remains contained within 1 °C, as calculated in our FEA. In any case, the study reports that temperatures of 46 °C and above caused tissue destruction in all animals, hemorrhage and paresis in some. No histologic changes were detected at temperatures up to 44 °C. These results suggest that the tolerable temperature of the spinal cord after local heating of 30 min was approximately 44 °C. Yamane et al. [41] conducted very similar research, which investigates the critical temperature of the spinal cord in hyperthermia produced by radiofrequency waves applied to the spine. The method was the same of that previously described, but they used rabbits as test subjects. After partial laminectomy, two electrodes and a thermosensor were introduced into the dorsal epidural space. Radiofrequency electrodes were placed on the skin of the back and the abdomen. Again, any change in spinal cord evoked potential (SCEP) reflects a change in the spinal cord function from an electrophysiological point of view. Thus, the focus was the measure of the SCEP amplitude. Their results revealed that a reduction in SCEP amplitude is correlated to a higher temperature test. They heated the spinal cord arbitrarily to 44 °C, waited a few minutes to allow it to warm completely and then turned off the RF system. While spinal cord returned to control temperature, SCEP was recorded at each reduction of 1 °C. At 44 °C and below, the amplitudes recovered to nearly the control value at the control temperature (37 °C, as our case). At 45 °C and above, the amplitudes decreased prominently or disappeared in the first several minutes and did not return to the control value at the control temperature.

As a result of this, the CUSA treatment seems to be safe, provided the application time at any single point remains brief, ideally not exceeding 1 s, to prevent localized overheating. For instance, the tip could vibrate once for 10 s at high amplitude (~60%), moving around to avoid focusing on the same application point for more than 1 s and preventing overheating of the area. It could then remain steady and perform only suction for several cycles, allowing the tissue to cool down before vibrating again. Alternatively, fragmentation could be alternated with other resection techniques that do not contribute to additional thermal impact. While our results show that even at 60% CUSA power, the temperature remains below the critical safety threshold, it’s crucial to note that further increases in power would likely lead to exceeding this limit, posing a risk of thermal damage to surrounding delicate tissues. Therefore, careful management of both application duration and power settings is essential for ensuring patient safety.

However, these conclusions must be considered in view of several limitations inherent to the proposed model and outlined below. First, the assumption that a spinal cord tumour presents a discrete and well-circumscribed mass, as represented in our model by a clearly defined region, is a simplification that does not accurately reflect the biological complexity of most Intramedullary Spinal Cord Tumours. Astrocytomas and many ependymomas, which are among the most common types encountered in clinical practice, often exhibit infiltrative growth patterns. These tumours blend irregularly with surrounding spinal cord tissue, making it difficult, if not impossible, to distinguish and separate tumour from healthy parenchyma either through microdissection or, even more so, via ultrasonic aspiration. Being these zones of infiltration extremely small in comparison with the main tumoral zone, the proposed model considers negligible their contribution to the heat and vibration transfer.

Second, the model assumes the persistent presence of anatomical structures such as epidural fat and cerebrospinal fluid (CSF) around the lesion site, which in reality are altered or removed during surgical exposure. Epidural fat is predominantly located dorsally and is typically resected during the surgical approach to the tumour, particularly in the posterior midline. Thus, its contribution as a thermal sink is likely overestimated. Similarly, CSF is continuously aspirated during tumour resection, which reduces its role in heat dissipation.

Finally, the model does not take into account the tumour’s vascularity and the dynamic changes in local perfusion during CUSA use. As the ultrasonic aspirator disrupts small vessels, the surgical field is often bathed in blood, which can significantly alter local thermal conditions. Blood itself may contribute to cooling the tissue, and the intact vasculature within and around the tumour can act as an endogenous heat sink, further mitigating temperature rises. As a consequence, the obtained results are a conservative evaluation of the heat transfer really occurring.

Taken together, the obtained results provide the basis for subsequent experimental validations. Future research could involve in vitro testing on tissue-mimicking phantoms specifically engineered to replicate the thermal conductivity, heat capacity, stiffness, and viscoelastic behaviour of biological tissues. These phantoms would allow for precise control and reproducibility of test conditions, enabling direct measurement of temperature profiles, vibrational propagation, and mechanical responses during CUSA application. The last step would include experimental validation on animal models.

Moreover, given the current absence of clear recommendations or evidence-based protocols for the use of CUSA in intramedullary tumour surgery, this work represents a first step toward defining safety thresholds and application strategies that could inform surgical guidelines in the future.

## 5. Conclusions

In this work, a Finite Element Analysis was performed to simulate vibratory and thermal interactions during CUSA application in microsurgical treatment of Intramedullary Spinal Cord Tumors. The specific aim was to identify any potential damage to the adjacent tissues, evaluating the thermos-mechanical effects during the utilization of the CUSA. A computational model of a vertebral column segment containing an IMSCT was developed and analyzed using ANSYS software. The simulations investigated strain distribution, heat generation, and temperature propagation within the biological tissues across two distinct simulation tests, varying the peak-to-peak amplitude of the displacement signal, i.e., 70 μm and 213 μm, with a frequency of 36 kHz for both tests.

The Finite Element Analysis (FEA) results provide insights into the biomechanical and thermal responses of tissue during Cavitron Ultrasonic Surgical Aspirator (CUSA) application. Specifically, the simulations consistently demonstrated that vibratory-induced strain remains highly localized to the immediate site of CUSA application. This localization is a crucial finding, indicating that mechanical stress is largely confined to the targeted tissue, thereby minimizing the risk of widespread mechanical damage to delicate adjacent structures, such as the spinal cord and surrounding neural elements.

Regarding thermal effects, the FEA model revealed that while temperature increases are measurable, they are consistently maintained below the critical safety threshold of 46 °C. This threshold is widely recognized in existing medical literature as the point beyond which irreversible tissue damage, particularly to neural tissue, can occur due to thermal necrosis. The maximum observed temperature increase in our simulations remained well within this safe limit, providing strong evidence for the thermal safety profile of CUSA when operated within the tested parameters. This outcome aligns favorably with current understanding of ultrasonic tissue interaction, suggesting that controlled CUSA application is unlikely to induce clinically significant thermal injury to neighboring tissues.

These conclusions contribute to the understanding of the thermo-mechanical interactions in ultrasonic tumour resection and aim to assist medical professionals in optimising surgical protocols. While it is evident that the safe and effective use of surgical tools such as CUSA depends primarily on the neurosurgeon’s clinical expertise, acquired through years of specialized training, the biomechanical modelling, here developed, can provide valuable insight into tissue behaviour under ultrasonic stress. Such a model provides the bases for a better understanding of the physical mechanisms involved which, in turn, may contribute to future experimental studies, support surgical planning tools, and contribute to improving patient safety in complex spinal procedures.

## Figures and Tables

**Figure 1 bioengineering-12-00842-f001:**
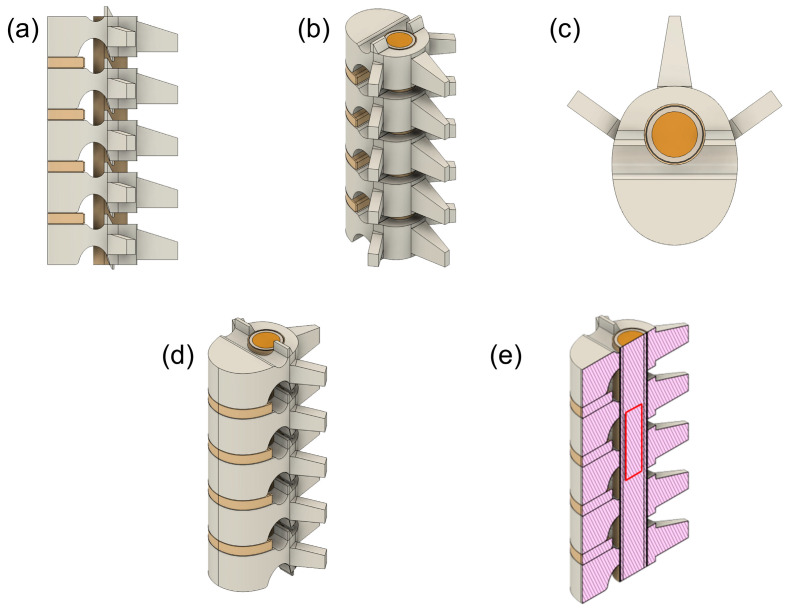
Human cervical model CAD. Sagittal view (**a**), posterior oblique view (**b**), axial (transverse) view (**c**), anterior oblique view (**d**), sagittal cross-sectional view with IMSCT modelling, outlined in red (**e**).

**Figure 2 bioengineering-12-00842-f002:**
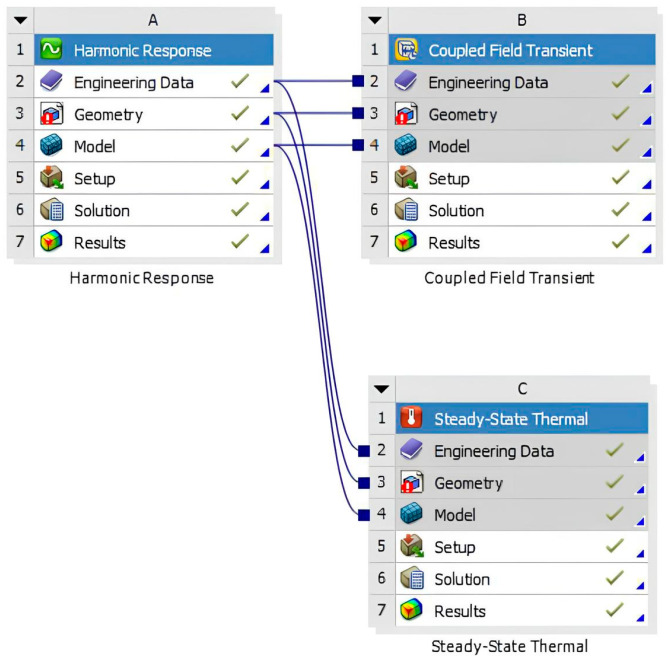
Ansys Workbench with three analysis blocks. Block (**A**) is the harmonic response analysis, block (**B**) is the coupled field transient analysis and block (**C**) is the steady-state thermal analysis. Connections between the blocks illustrate that block (**B**,**C**) receive the same input for their engineering data and geometry from block (**A**).

**Figure 3 bioengineering-12-00842-f003:**
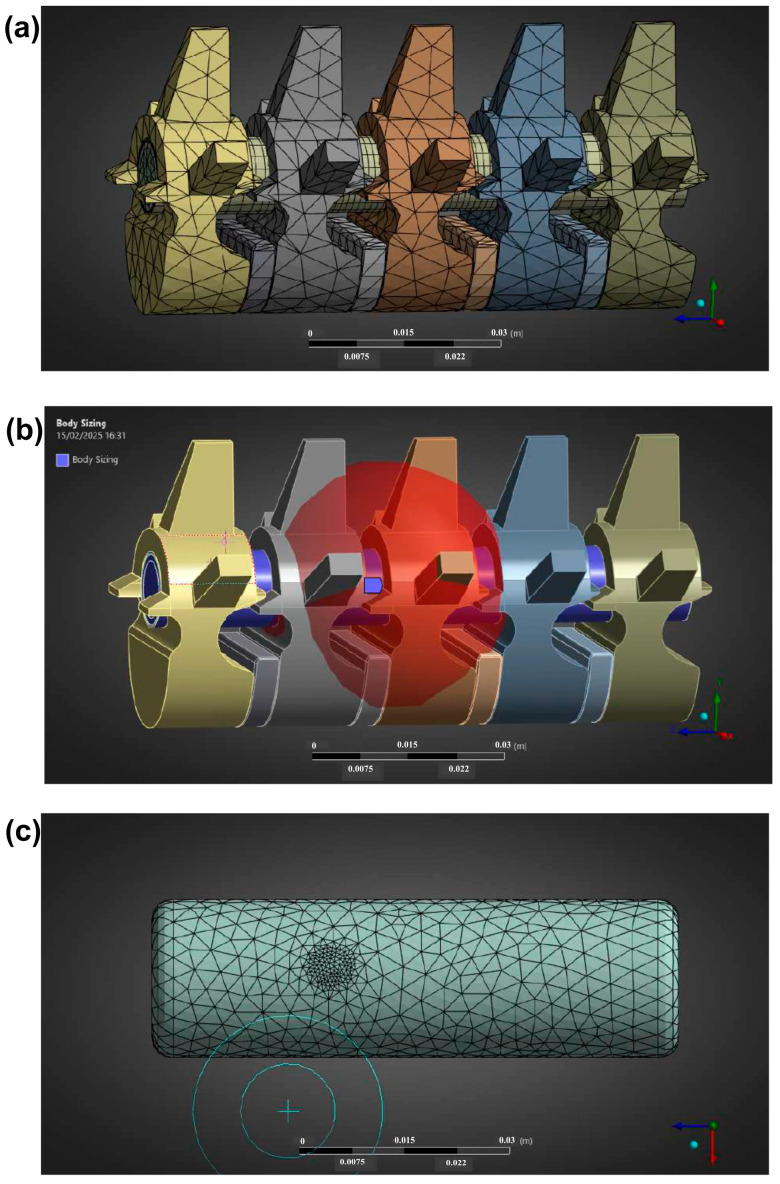
Resulting volumetric mesh of spinal cord model. Full body mesh (**a**), full body with sphere of influence (**b**) and tumour mesh and point of application of the device (**c**).

**Figure 4 bioengineering-12-00842-f004:**
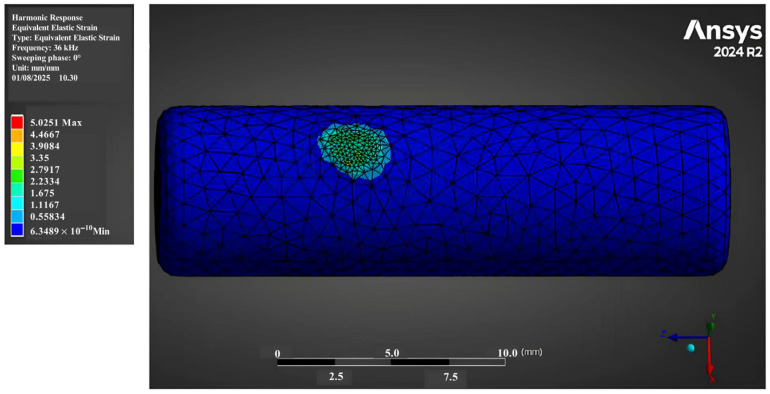
Strain field distributions on tumour when applying a displacement signal equal to 70 μm.

**Figure 5 bioengineering-12-00842-f005:**
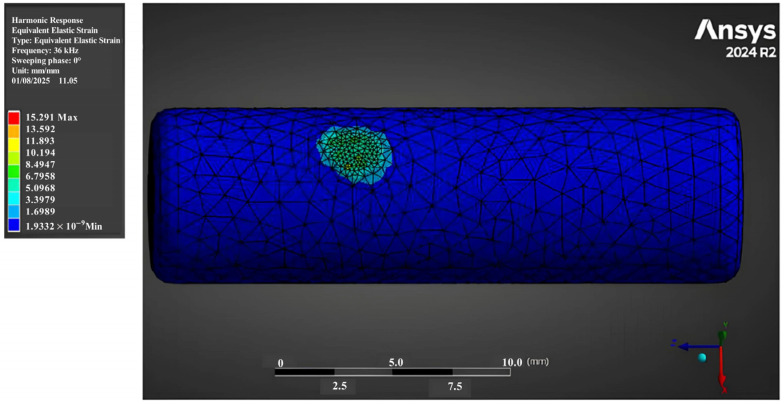
Strain field distributions on tumour when applying a displacement signal equal to 213 μm.

**Figure 6 bioengineering-12-00842-f006:**
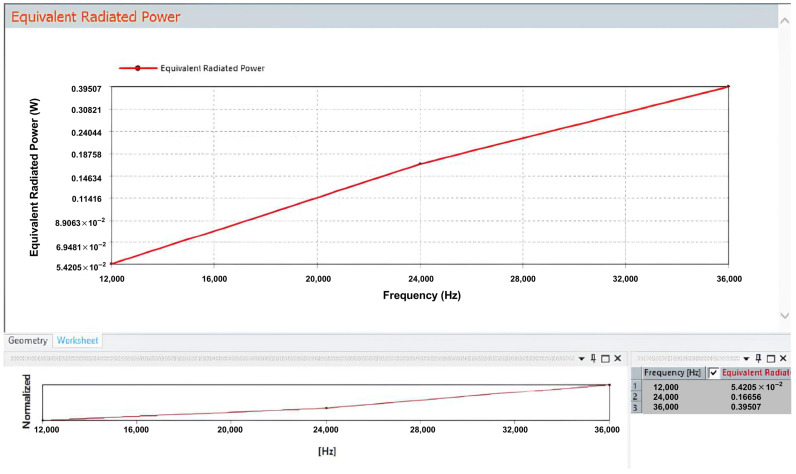
Equivalent radiated power by varying the frequencies when applying a displacement signal equal to 70 μm.

**Figure 7 bioengineering-12-00842-f007:**
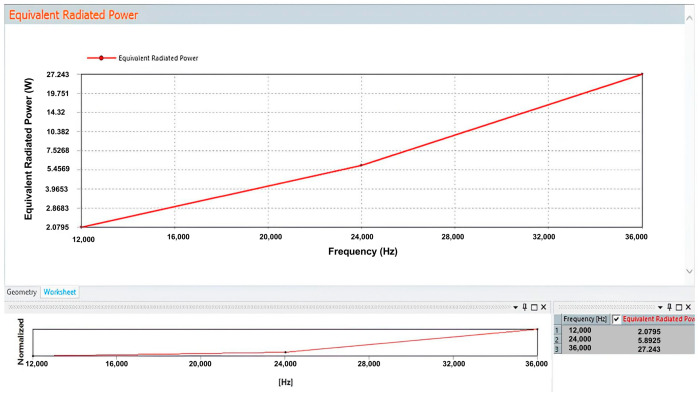
Equivalent radiated power by varying the frequencies when applying a displacement signal equal to 213 μm.

**Figure 8 bioengineering-12-00842-f008:**
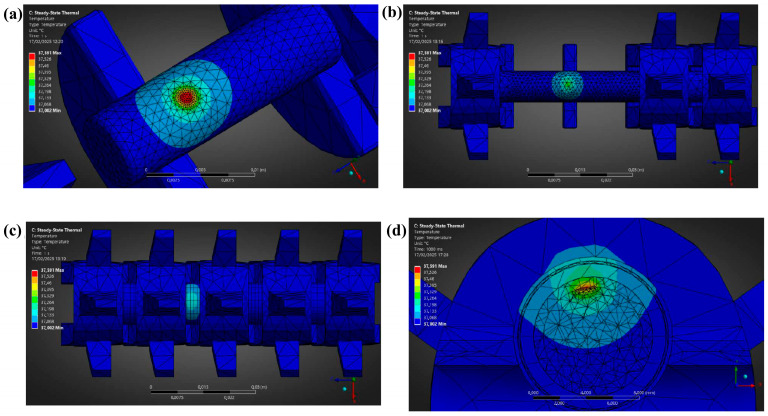
Temperature fields when applying a displacement signal of 70 μm on the different element of the model: (**a**) point of application of the signal, (**b**) spinal cord, (**c**) epidural fat, (**d**) cross sectional view at the tumour site.

**Figure 9 bioengineering-12-00842-f009:**
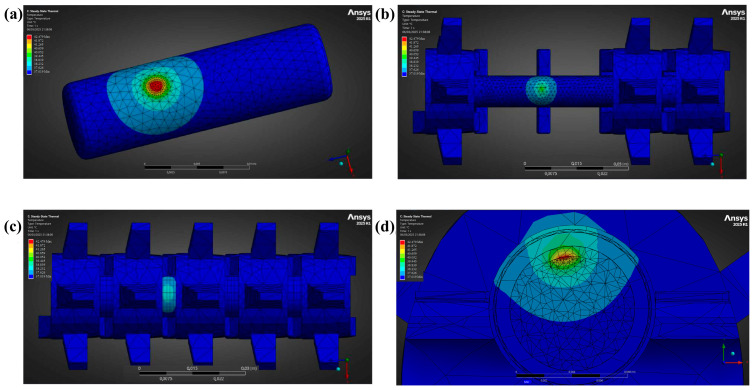
Temperature fields when applying a displacement signal of 213 μm on the different element of the model: (**a**) point of application of the signal, (**b**) spinal cord, (**c**) epidural fat, (**d**) cross sectional view at the tumor site.

**Table 1 bioengineering-12-00842-t001:** Geometrical dimensions of modelled parts.

Element	Measurement	Dimension (mm)
Overall Cervical C3–C7	total height	90
	body height	14
Vertebra	mediolateral width	24.5
	anteroposterior width	14
	cavity diameter	12
Intervertebral Disc	height	4
Epidural Fat	diameter	12
	thickness	0.3
Dura Mater	diameter	11.4
	thickness	0.2
CSF	diameter	11
	thickness	1
Spinal Cord	diameter	9
Tumour	diameter	7
	body height	25

**Table 2 bioengineering-12-00842-t002:** Elements properties.

Tissue	Density (g/cm^3^)	Young’s Modulus (MPa)	Poisson’s Ratio	Thermal Conductivity (W/m °C)	Heat Capacity (J/kg °C)
Bone	0.131 [27]	7600 [25]	0.3 [25]	0.68 [28]	1313 [29]
Intervertebral Disc	1.1 [18]	500 [25]	0.3 [25]	0.49 [18]	3568 [18]
Epidural Fat	0.911 [18]	3 [30]	0.49 [23]	0.2 [31]	2348 [29]
Dura Mater	1.174 [18]	31.5 [32]	0.3 [32]	0.44 [33]	3364 [33]
CSF	1 [26]	10^−3^ [26]	0.4887 [34]	0.57 [18]	4096 [18]
Spinal Cord	1.075 [18]	40.12 × 10^−3^ [35]	0.49 [25]	0.51 [18]	3630 [18]
Tumour (glioma)	1.075 [18]	1.8 × 10^−3^ [36]	0.3 [36]	0.5 [37]	3621 [37]

**Table 3 bioengineering-12-00842-t003:** Maximum temperature reached in the tissues of interest for the two performed simulation tests.

Element	Temperature (°C)	
	70 μm Displacement	213 μm Displacement
Tumour	37.59	42.48
Bone	37.07	37.62
Intervertebral Disc	37.02	37.02
Epidural Fat	37.18	38.75
Dura Mater	37.19	38.83
CSF	37.27	39.45
Spinal Cord	37.57	41.94

## Data Availability

The raw data supporting the conclusions of this article will be made available by the authors on request.

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
