# Peer review of "Ultrasonic Surgical Aspirator in Intramedullary Spinal Cord Tumours Treatment: A Simulation Study of Vibration and Temperature Field"

_bioengineering, 2025, doi:10.3390/bioengineering12080842_

Round 1
Reviewer 1 Report
Comments and Suggestions for Authors
General Description
This manuscript provides a detailed description of the CUSA’s size and vibratory and aspiration parameters and a nicely developed model of spinal cord tumor for ultrasonic dissection. The assumptions of length of time of vibration application, geometry of its application to the dissection surface and assumptions about the devices use on the exposed tumor are quite realistic. This model may be an important point of departure for assessment of circumstances that more realistically reproduce the surgical experience.
General Comments
Given the relatively common nature of surgical procedures on spinal cord tumors at most major medical centers, direct measurements of vibration and temperature parameters could probably be made in a clinical setting such as that which would be much more meaningful.
As the Cavitron Ultrasonic Surgical Aspirator has been in active use since the mid-1980’s I would dispute that it is innovative. Multiple similar, more dependable, less cumbersome devices by other manufacturers have been developed since then.
For those that use these devices regularly, the conclusion that “These findings suggest that CUSA can be safely used within defined operational parameters” provides no added information. The settings for intensity of vibration and rate of irrigation are easily understood and well known to surgeons who use the device regularly. Tolerance of the spinal cord to ultrasonic dissection is unique to each individual. Knowledge of the spinal cord tolerance to the use of the ultrasonic dissector is informed continuously with the use of spinal cord monitoring of motor and sensory pathways during the procedure and thus adjustments can be made in real time. A predetermined set of parameters may serve as a useful point of departure but are not to be considered as applicable beyond that point.
Overall, the conclusions fit the parameters of the model. However, they are not particularly of use to clinical practice.
Specific Comments
In figure one, assumptions that a spinal cord tumor is discrete, as in the red box, are a convenient but incorrect model of a spinal cord tumor. All astrocytomas and most ependymomas, the most common spinal cord tumors encountered, do not have a discreet edge. Rather the tumor infiltrates into the surrounding normal tissue, mixing with it in a manner that does not allow complete separation of the normal and tumor tissue with microdissection, let alone ultrasonic aspiration.
Line 149-150: please reword the sentence to remove the word “no” and better explain what is meant in relationship to discontinuities and mechanical properties.
Line 240: It is presumed “plied” is “applied”
Line 249: change “displacement” to “displacement”
Lines 254-256: This is an assumption that further limits applicability of this research in view of the remarkable heterogeneity of intrinsic spinal cord tumors. Astocytomas more often follow the assumption of being easy to fragment, but ependymomas can be remarkably firm, requiring sharp dissection to remove them in a piecemeal fashion.
Figure 4b and 5b add little to the information in this manuscript and could be dealt with in a simple sentence noting the structural components of the cervical spine are not impacted.
Figures 6 and 7 are a nice demonstration of the linear nature of radiated power across the tumor surface as frequency increases. The fallacy is, of course, is that there not a distinct tumor surface in astrocytomas and most ependymomas because the edges simply infiltrate in a relatively irregular manner into the surrounding normal spinal cord.
Lines 287-291: The epidural fat in the region of the tumor is removed during exposure of the lesion. The assumption that epidural fat is present uniformly around the circumference of the epidural space is incorrect. Most of it is dorsal, and therefore, as just mentioned, most of it is removed in the region of the tumor resection and cannot serve as a heat sink. The CSF is aspirated away continuously during the surgical resection of the tumor and therefore calculation/estimation of the heat dissipation into these structures during surgery is irrelevant.
Line 309-311: as mentioned above, little epidural fat is left for heat mitigation in the region of the tumor surgery.
Line 320-321: Exposure of the bone to heat (particularly the minor increases noted here) not of great interest, in view of the perspective that even autoclaved bone maintains relatively good structural integrity.
Lines 401-404: The technique suggestions seem reasonable but bring to mind a point completely missed in the model. The tumor has a blood supply and as the CUSA is used, vessels are disrupted causing the field to be bathed in varying amounts of blood, which will also dissipate heat. Also, the intact vasculature of the remaining tumor will also serve as a heat sink, reducing heat buildup. What makes it interesting from a surgeon’s standpoint is that each tumor has somewhat different degrees of vascularity. From an engineer’s standpoint, this variability adds an additional variable to minimize the value of a model.
Line 420: “themos” is like to be “thermos”
Line 429-432: this conclusion for the model is correct. However, for the target community, neurosurgeons using the CUSA for intrinsic spinal cord tumors, it is a statement of the obvious as this is well known base on intraoperative real time monitoring of spinal cord motor and sensory activity.
Reviewer 2 Report
Comments and Suggestions for Authors
This is a very good and original study on the vibration and temperature for use in the spinal cord. The authors should be commended on the study design.
Reviewer 3 Report
Comments and Suggestions for Authors
This is an interesting laboratory simulation of ultrasonic vibrations and temperature-effects on tissue surrounding the tip of the CUSA in a model of spinal cord tumors. It adds to the understanding of the CUSA-function regarding the heat flux in different settings. The authors conclude that its application in spinal cord intramedullary tumor surgery is safe when using brief applying times. But it remains a simulation: As they already mention in the discussion, the function of the spinal cord can fully recover if it is not heated above 44 °C. But temperature is not the only factor: The mechanical stress due to the vibration of the CUSA on the spinal cord can already damage and irritate the conduction of potentials. Temperature itself is-as mentioned in the manuscript-controlled by irrigation/cooling and should not be a problem. The mechanical stress is only local, but if it comes to the border between tumor and spinal cord tissue – may cause damage. Therefore, defining the tumor borders remains crucial in operating spinal cord tumors.
To prove their hypothesis and possible clinical impact of the model, some measurements at least in cadaver studies or better with functional analysis in animal models should be performed in a next step.
Some minor issues:
Introduction:
Reference (10) is listed as Suetsuna, F. et al., in the text the authors mention Kirchner et al. which I cannot find in the reference list. Please correct.
Materials and methods:
Figure 1: Please use the conventional radiological terms: Sagittal, axial view etc.
Figures 2-9: Images of poor quality, legends (on the images) too small and therefore not readable
Round 2
Reviewer 1 Report
Comments and Suggestions for Authors
The authors have certainly made a reasonable attempt to address the criticisms of the first review. There are some comments worth making here:
• For the response stating “there are no established, guidelines, recommendations…”: It is correct there no written guidelines as to device settings for spinal cord tumors. However, the nature of a multi-year neurosurgery residency imparts this knowledge to neurosurgeons in a practical and easy to understand fashion. It is true, and frustrating for engineers, that there is fluid and difficult to quantify indications and methods for use of many surgical tools, including the CUSA. That is part of the art of neurosurgery. In fact, publishing specific settings and techniques that are then followed by inexperienced surgeons as valid, may actually leave the authors of this manuscript and the surgeons that use the guidance open to medical litigation if they are used and a spinal cord injury is incurred.
As far as references 14-16, which the authors deem important, the following should be noted:
• Reference 16 is an abbreviated review of spinal cord tumors, noting the challenges in their management. But it does not suggest or review any models, such as the one in the manuscript being reviewed here, for their management.
• Reference 14 and 15: These references deal with a model to test the integrity of artificial muscle in vitro. It is true some geometric assumptions are made in these two references that seem valid. However, the simplistic structure of muscle verses the elegant nature of the spinal cord does not lend itself to these assumptions.
The additions to “Discussions and Model Limitations” section are useful in terms of being honest about the model. Having added those comments, the overall impact of the manuscript is realistically diminished. This is because there are so many shortcomings that the manuscript, or any downstream work, has little chance of leading to any contribution to management of spinal cord tumors.
The new paragraphs in the revision require English editing
Reviewer 3 Report
Comments and Suggestions for Authors
With some additions in introduction and discussion regarding model properties, future research and possible impact on CUSA-application, the authors added valuable thoughts to their publication.
Also the figures are of better quality and add to the understanding of the research.
